# The Role of Intention, Behavioral Regulation, and Physical Activity Behavior in the Prediction of Physical Activity Identity across Time

**DOI:** 10.3390/bs14100886

**Published:** 2024-10-01

**Authors:** Colin M. Wierts, Edward Kroc, Ryan E. Rhodes

**Affiliations:** 1School of Exercise Science, Physical and Health Education, University of Victoria, 3800 Finnerty Road, Victoria, BC V8P 5C2, Canada; rhodes@uvic.ca; 2Measurement, Evaluation, & Research Methodology Program, University of British Columbia, 2125 Main Mall, Vancouver, BC V6T 1Z4, Canada; ed.kroc@ubc.ca

**Keywords:** multi-process action control, self-regulation, exercise, integrated regulation, longitudinal design, behavior change, maintenance

## Abstract

Physical activity identity represents an important determinant of sustained physical activity behavior. The purpose of this investigation was to examine whether intention, behavioral regulation, and moderate-to-vigorous physical activity (MVPA) behavior explain significant variation in physical activity identity across time. Using a repeated measures observational design, lower-active adults new or returning to physical activity participation (*N* = 66) completed measures of study variables every three weeks over the course of a nine-week period (four assessments total). Based on the results of mixed-effects regression modelling, there was a small, non-significant increase in physical activity identity across time (*b* = 0.07, *p* = 0.13). Intention, MVPA, and behavioral regulation mostly had significant (*p*s < 0.05) bivariate correlations with physical activity identity at the same time point of assessment. Behavioral regulation explained significant variation in physical activity identity across time (*b* = 0.26, *p* < 0.0001), but intention and MVPA were non-significant (*p*s > 0.05) after including a random intercept and controlling for behavioral regulation. Identity was resistant to change among new physical activity initiates in this study and longer time frames of assessment are needed (e.g., six months). Behavioral regulation should be examined as a determinant of physical activity identity in future investigations.

## 1. Introduction

The health benefits of participating in moderate-to-vigorous physical activity (MVPA) behavior are well established and include (but are not limited to) a reduction in early onset mortality [1], reductions in a number of chronic health conditions including several forms of cancer [1], and improved mental health [2] and well-being [3]. However, it has been estimated that 27.5% of adults worldwide are insufficiently active (i.e., participating in less than 150 min of MVPA per week), and this prevalence of inactivity is much larger in high income countries [4]. Therefore, understanding why (or why not) individuals engage in physical activity behavior is a relevant research endeavor (e.g., [5,6,7,8,9,10]).

Although understanding physical activity engagement is important, researchers have also highlighted the importance of understanding why individuals maintain their physical activity behavior over the long term, after initial engagement [11,12,13,14]. However, many of the theoretical frameworks applied to physical activity behavior, including social cognitive models (e.g., theory of planned behavior [15]), humanistic models (e.g., self-determination theory [16]), and some dual process models (e.g., affective-reflective theory of physical activity [6]), make no distinction between the constructs involved in the initiation and maintenance of physical activity behavior [11]. More recently, researchers have highlighted the likely importance of identity [11,17,18], and the related concept integrated regulation [11,17], for the understanding of physical activity and behavior change maintenance. 

Identities refer to the various components that encompass the self-concept or sense of self (i.e., who am I?), and can include culturally relevant personal characteristics like interests and attributes (person or self-identities), formal social roles such as ‘parent’ or ‘student’ (role identities), and relevant group memberships like ‘us Lakers fans’ or ‘us Park Runners’ (social identities) [19]. Physical activity (exercise) identity refers to the internalization of physical activity and exercise behavior as an important part of the self-concept (“I am a physically active person”) [20]. A conceptually similar construct to physical activity identity is integrated regulation, a self-determined form of motivation whereby individuals participate in physical activity because it aligns with their sense of self and values [16].

Identities (or integrated regulations) are an important source of motivation for behavior as they represent internalized standards for behavior across various contexts [21]. When behavior matches the identity standard it promotes positive affect and identity verification. When there is a discrepancy between behavior and the identity standard the result is negative affect, which motivates behavior modification to match the identity standard [19,22]. Therefore, identities support cognitive-affective feedback mechanisms that lead to long-term behavioral maintenance [19,21,22]. Indeed, there is a substantial body of research that suggests when individuals identify as a physically active person or as an exerciser, they are more likely to engage in physical activity behavior [20]. Moreover, it has been demonstrated in past research that physical activity identity is the most salient predictor of physical activity behavior after controlling for many other relevant psychological predictors [23,24].

Although physical activity identity has been established as an important determinant of physical activity behavior, less attention has been devoted to understanding the antecedents of identity development across time [20]. Therefore, understanding the factors that predict physical activity identity and its development is an important area of investigation, particularly for physical activity intervention development. In their systematic review of physical activity identity, Rhodes et al. [20] examined relevant correlates of identity among fifteen studies. The authors [20] determined that commitment to physical activity and perceived capability were relevant correlates of identity. This is in line with recent evidence [23,25] and theory [26] that suggests perceived ability and competence are relevant predictors of physical activity identity. Rhodes and colleagues [20] also reported that affective judgements (outcome expectations, attitudes, intrinsic motivation), autonomous motivation, and social activation variables (relatedness, social support) were relevant correlates of physical activity identity. A shortcoming of current research however, as noted in this review [20], was that a majority (n = 13) of the studies utilized cross-sectional designs. It stands to reason that the current state of evidence regarding antecedents of physical activity identity provides little indication of how physical activity identity may develop across time. 

To date, there have been few investigations examining predictors of physical activity identity (or identity change) across time. Ntoumanis et al. [27] examined motivational predictors, operationalized within self-determination theory [16], of change in exercise identity over three time points within a six-month time period among regular exercisers who belonged to fitness centers. None of the motivational predictors (intrinsic motivation, identified regulation, introjected regulation, external regulation, and amotivation) explained significant variation in change in exercise identity (i.e., change in slope). However, intrinsic motivation (i.e., exercising for enjoyment purposes) predicted variation in exercise identity across time [27]. A limitation of Ntoumanis et al. [27] was that the sample consisted of active individuals whose exercise identity was likely to be stable across time as identity is considered to be resistant to change [22]. More recently, Wierts et al. [28] examined changes in running identity among individuals enrolled in the Run to Quit program, a 10-week multiple heath behavior-change program targeting running (i.e., exercise) behavior and smoking cessation. Changes in running self-efficacy was the most important predictor of changes in running identity, followed by changes in running behavior (i.e., number of runs per week), and attractions (i.e., enjoyment and satisfaction) to the group exercise tasks (i.e., task cohesion) [28]. A strength of Wierts et al. [28] was that the authors were able to examine change in running identity among lower-active individuals (i.e., individuals primarily new to running). However, a limitation was that the authors utilized secondary data from a program evaluation and chose potential predictors of running identity based on various identity theories [28]; therefore, lacking one coherent theoretical approach for examining change in physical activity identity. Another limitation of Wierts et al. was that they utilized residualized change scores between two time points (baseline to follow-up/post-program), which did not provide an opportunity to examine the pattern of change in running identity across the program.

One theoretical model that may be particularly useful for making predictions about physical activity identity change among initially low-active individuals is the multi-process action control (M-PAC; [29]) framework, a meta-theory of physical activity behavior change and maintenance. The M-PAC model specifies that initial physical activity behavior change is determined primarily by strong intentions to be physically active and successful behavioral regulation strategies such as action planning and behavioral monitoring [29]. Reflexive processes (identities and habits) are expected to develop over time as a result of learned associations with physical activity and reflecting on the successful translation of intention into behavioral experience [29]. Thus, habits and identities are theorized to support long-term maintenance of physical activity [29]. Based on the M-PAC framework, intention, behavioral regulation, and action control (i.e., physical activity behavior) are key determinants of change in physical activity identity [29]. 

The M-PAC model is also aligned with existing research and other theoretical frameworks detailing the nature of physical activity identity and its development. Specifically, authors of identity theories [19,21,22,30] highlight the importance of successful behavioral enactment in the development of identity. For example, the physical activity self-definition model (PASD; [26]) operationalizes ‘perceived wanting’ and ‘perceived commitment’, which are conceptually similar to intention, as determinants of physical activity identity. The PASD [26] also operationalizes ‘perceived trying’, which is conceptually similar to behavioral regulation, as a determinant of physical activity identity. In line with PRIME theory [31], behavioral regulation may assist in setting rules for oneself (e.g., through plans) which could foster identity. Work by Strachan and colleagues [32,33,34] has also linked intention, behavioral regulation, and physical activity behavior to physical activity identity. Strachan et al. [32] qualitatively examined the meanings people attribute to their exercise identity. Participants reported that ‘exercisers’ (i.e., people with an exercise identity) engage in goal-directed and purposeful exercise [32], which aligns with having strong physical activity intentions. Participants also reported that ‘exercisers’ prioritize exercise and engage in consistent exercise [32], which aligns with behavioral regulation and physical activity behavior as predictors of physical activity identity. Finally, Strachan et al. also reported a strong correlation between both self-regulatory efficacy [33] and behavioral regulation [34] with physical activity identity. In summary, there are theoretical and empirical links between physical activity identity and behavioral regulation, intention, and physical activity behavior. However, there is a lack of empirical testing of the relationship between these three variables and identity among people new or returning to physical activity behavior. 

Thus, the purpose of this study was to examine (a) whether physical activity identity changes across time among low-active individuals that are new or returning to physical activity and (b) whether M-PAC variables (intention, behavioral regulation, and physical activity behavior) predict physical activity identity across time and change in physical activity identity. It was hypothesized that physical activity identity would significantly increase across the nine-week period of the study, based upon the fact that the sample included new exercise initiates intending to improve upon their physical activity behavior [19,21,22,29,30]. It was also hypothesized that intention, behavioral regulation, and MVPA would explain significant variation in physical activity identity across time and significant variation in change in physical activity identity [29]. 

## 2. Methods

### 2.1. Participants and Procedures

This study received ethical approval from the Human Research Ethics Board at the University of Victoria on 16 July 2019 (protocol number: 18-1208). Seventy-five adults (19+ years of age) were recruited from fitness/recreational centers and on social media (Facebook, Instagram) in a medium sized city in Western Canada. All participants provided informed consent before participating. In order to be eligible for the study, participants needed to self-identify as new/returning exercisers (recently decided to start exercising regularly or had only started exercising regularly less than two weeks prior to study enrollment) and report not meeting the Canadian physical activity guidelines (engaging in less than 150 min of MVPA [35]) for six months prior to study enrollment. Using a prospective observational design, participants were asked, via email, to complete an online self-report survey (via SurveyMonkey) every three weeks over the course of a nine-week period (four surveys total). 

### 2.2. Measures

#### 2.2.1. Physical Activity Identity 

Physical activity identity was measured using the four-item integrated regulation subscale [36]. The four items tap into the degree to which people exercise because it aligns with their identity (e.g., “I consider exercise part of my identity”). Participants responded to the four items (see items in Appendix A) on a five-point scale ranging from 0 (Not true for me) to 4 (Very true for me). Each item was prefaced with the statement, “Why do you engage in exercise”? Although researchers generally use the exercise identity scale [37] or the role identity subscale [38] within the exercise identity scale for the assessment of physical activity identity, the integrated regulation subscale shares high conceptual similarity with the exercise identity scale [37] and the two scales are highly correlated, *r* = 0.82 [34].

#### 2.2.2. Physical Activity Intention

Physical activity intention was operationalized as intention strength (i.e., desire to perform MVPA; [39]) and was measured using a single item [40]: “My goal is to engage in moderate or vigorous exercise for at least 150 min per week over the next month”. Intention was operationalized as intention strength as opposed to decisional intention (yes–no dichotomy) as intention was theorized, in the current investigation, to be the culmination of reflective beliefs (capability, opportunity, attitudes) towards physical activity behavior. Participants rated their agreement with the statement on a seven-point scale ranging from Strongly Disagree (−3) to Strongly Agree (3).

#### 2.2.3. Behavioral Regulation

Regulation of physical activity behavior was assessed using four items (see items in Appendix A) adapted from Sniehotta et al. [41] and Umstattd et al. [42]. The four items tapped into the extent that participants planned and self-monitored their physical activity behavior (e.g., “I kept track of my physical activity in a diary or log over the last 2 weeks”). Participants responded to the four items using a seven-point scale ranging from Strongly Disagree (−3) to Strongly Agree (3).

#### 2.2.4. Physical Activity Behavior

Physical activity behavior (i.e., action control) was operationalized as MVPA, which was measured via the Godin leisure time exercise questionnaire [43,44]. Participants were asked to report the number of times (i.e., frequency) they engaged in mild (i.e., minimal effort, no perspiration), moderate (i.e., not exhausting, light perspiration), and vigorous (i.e., heart beats rapidly, sweating) physical activities in the prior week, as well as the number of minutes (on average) they engaged in those activities. A score for MVPA was calculated using the following formula [44]: moderate frequency × average moderate duration + vigorous frequency × average vigorous duration

### 2.3. Analysis

Descriptive statistics, reliability statistics, and bivariate correlations between variables were calculated using SPSS version 27. The main analysis (fixed- and mixed-effects regression modelling) was conducted using R software (version 4.40; [45]). Several R packages were used, including the stats package [45] for standard fixed-effects modelling, the linear and nonlinear mixed-effects models (nlme) [46,47] package for mixed-effects modelling, and lattice [48] and latticeExtra [49] for examining fitted vs. residual plots. The data and R code for the main analysis can be found at OSF (https://osf.io/qu6gf/ (accessed on 10 June 2024)). 

Fixed- and mixed-effects (i.e., multilevel) regression modelling was used to examine physical activity identity changes across time, the relationship between M-PAC variables (intention, behavioral regulation, physical activity) and physical activity identity across time, and whether M-PAC variables explain variation in change in physical activity identity (i.e., change in time slope). Initial fixed-effects regression models were used to examine the relationships between predictors (time, M-PAC variables) and physical activity identity. Given that participants completed multiple assessments across time (i.e., time was nested within each participant), mixed-effects models with random intercepts were created to account for differences in baseline physical activity identity (i.e., different starting points in the outcome variable). When incorporating random intercepts an autoregressive covariance structure was utilized which assumes that for the outcome variable (i.e., physical activity identity) correlations are largest between adjacent time points and become smaller between farther time points [50].

Sequential time models were built to determine whether physical activity identity changed in a linear or quadratic fashion across time. The time models included (a) a linear time slope (model 0), (b) inclusion of a quadratic slope (model 1), and (c) inclusion of a random intercept but no quadratic slope as it did not improve model fit (model 2). Next, separate models were sequentially built for each predictor variable (intention, behavioral regulation, and MVPA), which included (a) a linear time slope and the predictor (a quadratic time slope was determined to be unnecessary) (model 0), (b) inclusion of a random intercept (model 1), and (c) inclusion of a time × predictor interaction (model 2) to determine whether each M-PAC variable was significantly related to identity across time, whether a random intercept was necessary to improve model fit, and whether each M-PAC variable predicted change in physical activity identity, respectively. The same sequential models were built for demographic variables, including age, gender (man or woman), and minority status (yes or no), to explore whether demographics were related to physical activity identity across time and change in physical activity identity. A final model was built which included a linear time effect, each predictor (intention, behavioral regulation, and MVPA), and a random intercept to determine the most relevant predictors of physical activity identity across time. The final model did not include time × predictor effects as none of the predictors explained significant variation in the physical activity identity time slope. An exploratory model was also built to control for demographic variables, and included a linear time slope, each M-PAC predictor (intention, behavioral regulation, and MVPA), demographic variables (age, gender, and minority status), and a random intercept.

Model validity (i.e., validity of the model residuals’ structure) was determined by examining fitted versus residual plots and model fit was assessed via the Akaike information criterion (AIC) and the Bayesian information criterion (BIC). Well-behaved residuals (including a null relationship with fitted values, constant spread, and lack of clustering/autocorrelation) suggest adequate model validity [51]. Lower AIC and BIC values correspond to better model fit; i.e., improved variance explained while guarding against overfitting [50]. Intraclass correlation coefficients, which corresponded to within-person effects, were also calculated when models included random intercepts. All available data from participants were used in the analysis, with no missing data imputed or otherwise filled in. 

## 3. Results

Among the 75 individuals recruited for the study, 66 participants (*M*_age_ = 34.47 years, *SD*_age_ = 9.50 years) completed at least one assessment of physical activity identity and were included in the sample for this investigation. See Table 1 for the complete sample demographics. See Appendix A for the descriptive statistics, bivariate correlations, and Cronbach’s alphas (for multi-item scales) for the study variables. There was a total of 200 observations for physical activity identity (average of three measures completed per participant across four time points) and 65 complete cases (i.e., completed measures for identity, intention, behavioral regulation, and MVPA) at baseline, 48 complete cases at week 3, 36 complete cases at week 6, and 33 complete cases at week 9. Therefore, 182 observations of a possible 264 were included in the final model (69%). Across the nine weeks, there was a small average increase in physical activity identity from baseline (*M* = 2.65, *SD* = 0.83) and week 3 (*M* = 2.51, *SD* = 1.01) to week 6 (*M* = 2.74, *SD* = 0.91), and to week 9 (*M* = 2.93, *SD* = 0.77). There was also an average increase in MVPA from baseline (*M* = 107.20, *SD* = 96.27) to week 3 (*M* = 124.48, *SD* = 93.02) and week 6 (*M* = 115.25, *SD* = 95.33), and a further increase to week 9 (*M* = 147.50, *SD* = 77.20), suggesting that, on average, participants increased their physical activity behavior and physical activity identity (to a small extent) across the nine weeks. The correlations between behavioral regulation and physical activity identity, when measured at the same time point, were moderate in strength and significant at baseline (*r* = 0.59, *p* < 0.0001), week 3 (*r* = 0.49, *p* < 0.001), week 6 (*r* = 0.59, *p* < 0.001), and week 9 (*r* = 0.35, *p* < 0.05). The correlations between MVPA and physical activity identity, when measured at the same time point, were mostly small–moderate and significant at baseline (*r* = 0.26, *p* < 0.05), week 3 (*r* = 0.37, *p* < 0.01), and week 6 (*r* = 0.39, *p* < 0.05). Interestingly, at week 9, the correlation between MVPA and physical activity identity was null (*r* = −0.04, *p* = 0.85). The correlations between intention and physical activity identity, when measured at the same time point, were mostly small–moderate and significant at baseline (*r* = 0.29, *p* < 0.05), week 6 (*r* = 0.39, *p* < 0.05), and week 9 (*r* = 0.41, *p* < 0.05). The correlation between intention and behavioral regulation at week 3 was null (*r* = 0.01, *p* = 0.96).

### 3.1. Time Models

There was a small, non-statistically significant linear increase in physical activity identity across time (*b* = 0.07, SE = 0.04, *p* = 0.13) after incorporating a random participant-specific intercept. Adding a quadratic time effect (*b* = 0.08, SE = 0.06, *p* = 0.18) did not improve model fit, suggesting that including a quadratic time effect was not necessary and that a linear time trend was most appropriate. See Appendix A for the time model coefficients. 

### 3.2. Intention Models

Physical activity intention was significantly related to identity across time (*b* = 0.20, SE = 0.05, *p* < 0.001). Incorporating a random intercept improved model fit, however, the relationship between intention and identity was no longer significant (*b* = 0.05, SE = 0.05, *p* = 0.27). Moreover, incorporating a time × intention effect (*b* = 0.01, SE = 0.04, *p* = 0.87) did not improve model fit and the coefficient was not statistically significant, suggesting that intention did not help explain variation in change in identity. See Appendix A for the coefficients corresponding to the intention models. 

### 3.3. Behavioral Regulation Models

Behavioral regulation was significantly related to physical activity identity across time (*b* = 0.38, SE = 0.05, *p* < 0.0001). Incorporating a random intercept improved model fit and behavioral regulation was still significantly related to identity (*b* = 0.26, SE = 0.04, *p* < 0.0001). However, there was no significant time × behavioral regulation interaction effect (*b* = −0.02, SE = 0.04, *p* = 0.59) suggesting that behavioral regulation was not associated with change in identity. See Appendix A for the coefficients corresponding to the behavioral regulation models. 

### 3.4. MVPA Models

MVPA was significantly related to identity across time (*b* = 0.003, SE = 0.0007, *p* < 0.001). Incorporating a random intercept improved model fit, however, the relationship between MVPA and identity was no longer significant (*b* = 0.001, SE = 0.0006, *p* = 0.42). The time × MVPA interaction effect (*b* = 0.00004, SE = 0.0005, *p* = 0.94) was also non-significant and did not improve model fit, suggesting MVPA did not predict change in identity. See Appendix A for the coefficients corresponding to the MVPA models. 

### 3.5. Final Model

The final model included a linear time effect and each of the M-PAC predictors (intention, behavioral regulation, and MVPA). None of the M-PAC variables significantly interacted with time (nor did any of the interaction effects improve model fit), and therefore no interaction effects between time and M-PAC variables were included in the final model. A random intercept was also included in the final model. With respect to the model coefficients, behavioral regulation was significantly related to physical activity identity across time (*b* = 0.26, SE = 0.05, *p* < 0.0001). However, intention (*b* = 0.002, SE = 0.05, *p* = 0.96) and MVPA (*b* = −0.0002, SE = 0.0006, *p* = 0.79) were not significantly related to physical activity identity across time. See Table 2 for the final model coefficients. Finally, none of the demographic variables (age, gender, and minority status) explained significant variation in physical activity identity across time or change in identity. Furthermore, the relationship between behavioral regulation and physical activity identity was still significant (*b* = 0.23, SE = 0.05, *p* < 0.0001) after controlling for the three demographic variables (See Appendix A). 

## 4. Discussion

Overall, the purpose of this investigation was to examine the role of intention, behavioral regulation, and MVPA in explaining variation in physical activity identity across time. We aimed to advance the current understanding of identity development in physical activity contexts by sampling individuals who were new or returning to physical activity participation. The sample used in this investigation is a prime population to study potential physical activity identity change as behavior is considered a pre-requisite of identity formation [19,21,22], and therefore participants in the sample had an opportunity to meaningfully change their identity across the nine-week sample period. We hypothesized that physical activity identity would increase across time and intention, behavioral regulation, and MVPA would explain significant variation in both identity across time and change in identity. 

On average, there was a small, non-significant, increase in physical activity identity across the study, and therefore we failed to support the hypothesis that identity would significantly increase across the nine-week period. In conjunction with the lack of change in identity, we also failed to support the hypothesis that intention, behavioral regulation, and MVPA would explain significant variation in change in identity. The lack of change in identity is in line with theory [22] whereby identities are theorized to be resistant to change. 

The lack of change in identity was potentially due to a lack of change in physical activity behavior [19,21,22] as there was a small average increase in physical activity behavior across the nine weeks of the study. Behavior is considered a necessity of identity development [19,21,22,30] as identities are essentially behavioral standards for action across various contexts [19,21,22]. Participants may have lost interest in following through with their physical activity intentions [52], a phenomenon also seen in interventions whereby participants are unable to sustain their physical activity behavior post-intervention [14]. Future investigations examining change in physical activity identity likely need larger samples, and perhaps more importantly, a greater number of repeated assessments of identity over a longer period of time in order to identify reliable patterns of identity change. Utilizing short form measures and limiting the number of variables included in each assessment could potentially decrease participant burden and dropout and retain the intended sample size. Future investigations studying identity change may also benefit from including intervention strategies to promote behavior and identity. Researchers could also include secondary measures of identity in physical activity interventions or program evaluations (e.g., [28]) to understand the nature of physical activity identity change.

With respect to the hypotheses regarding predictors of identity across time, the hypothesis that behavioral regulation was significantly related to physical activity identity across the nine-week study period was supported. The significant relationship between behavioral regulation and identity is in line with existing evidence [32,33,34]. For example, there is qualitative evidence that individuals with exercise identities prioritize exercise over other activities and create exercise plans and schedules that require discipline and organization of one’s time to effectively execute [32]. Another major theme of Strachan et al.’s [32] qualitative investigation was that people who are exercisers engage in consistent exercise. The theme of consistency is also in line with behavioral regulation as behavioral consistency requires effective planning and scheduling of one’s time. 

The link between behavioral regulation and physical activity identity is also in line with past observational research [33,34]. In this investigation, physical activity identity was the dependent variable, whereas Strachan and colleagues [33,34] operationalized behavioral regulation as an outcome of having a strong physical activity identity. The M-PAC [29] framework does place identity at the apex of the psychological processes involved in physical activity behavior and maintenance and specifies that identity develops after behavioral regulation strategies have been effectively utilized. However, the M-PAC [29] framework also allows for psychological determinants of physical activity behavior to have reciprocal relationships across time [29]. Specifically, individuals with physical activity identities are expected to practice behavioral regulation strategies in order to continuously verify their identity [29]. Based on the theorizing within the M-PAC framework [29], it is likely that behavioral regulation and identity are reciprocally related, such that successful behavioral regulation promotes identity verification, and individuals with a physical activity identity will use behavioral regulation strategies to support identity verification. It should be noted that we focused on the behavioral components of self-regulation (planning, scheduling, and self-monitoring) in this investigation. However, it is apparent that individuals with an exercise identity also practice emotional regulation in order to overcome undesirable feelings, such as fatigue, to follow through with their physical activity intentions [32]. Future research could examine the relationship between the emotional component of self-regulation (i.e., reactive regulation; [53,54]) and physical activity identity across time. Future research could also test the effects of behavioral regulation interventions on physical activity identity to determine whether behavioral regulation strategies precede identity change. 

In contrast to behavioral regulation, the hypothesis that intention and MVPA would be significantly related to identity across time was not supported. The lack of variance in identity explained by intention and physical activity is potentially due to low variability in intention and MVPA. Specifically, many of the participants in the current investigation were intenders, as recruitment criteria specifically sought participants who were interested in initiating MVPA. The sample also included lower-active individuals not meeting the physical activity guidelines six months prior to study enrollment, as per the inclusion criteria of the study. With this in mind, the participants in the current investigation were likely too similar in their MVPA behavior and physical activity intentions to see significant relationships between these variables and physical activity identity. The reduction in effect size for intention and MVPA after the inclusion of random intercepts may have also been attributable to the unbalanced nature of the data (i.e., incomplete responses at follow-up time points). Future investigations examining identity promotion will likely be conducted with populations similar to this sample where participants are exercise initiates with high intentions. It stands to reason then that MVPA and intention are not the clear markers for understanding identity among similar samples, however, future investigations are needed to support this finding. 

The findings of this investigation, particularly the strong link between behavioral regulation and physical activity identity, have practical implications for health practitioners looking to support their clients’ physical activity behavior. Health practitioners are well positioned to support their clients’ physical activity behavior via identity development [20,29]. Health practitioners could help their clients develop behavioral regulation skills to potentially support their physical activity identity development [29]. One relevant behavioral regulation tactic is the establishment of a detailed plan regarding when, where, and how one will do physical activity, as well as strategies for overcoming barriers to physical activity and prioritizing physical activity over other behaviors [29,55]. Another key behavioral regulation skill is to monitor the progress one makes towards meeting their physical activity goals and intentions [29,55]. By developing behavioral regulation skills, individuals can actively reflect on their successful translation of intention into action and learn to prioritize physical activity over other behaviors, both of which are likely important for physical activity identity development [29]. 

Despite the novel theoretical test of physical activity identity development and highly relevant sample to understanding PA promotion, this study has limitations that should be considered when interpreting the findings. Our study was limited by the short time frame of assessment (nine weeks) and the number of repeated assessments (four). The limited number of repeated assessments likely affected our ability to detect patterns of change in identity and should be considered in future investigations of identity and physical activity behavior change. There were also limitations in the measures used in this investigation. We chose to use the integrated regulation subscale [36] to operationalize physical activity identity. The exercise identity scale [37] or the role identity subscale [38] within the exercise identity scale are more regularly used to assess identity in physical activity settings and future research should utilize these scales to confirm our findings. With respect to measuring behavioral regulation, we used items adapted from Sniehotta et al. [41] and Umstattd et al. [42]. However, a more refined measure of behavioral regulation with validity evidence has been recently developed (physical activity regulation scale; [53]) and could be used in future investigations examining identity across time. We also relied on self-report assessments of physical activity behavior which may have been affected by social desirability and recall biases and future investigations may consider utilizing direct assessments of physical activity such as accelerometers. Finally, our sample was limited to primarily younger adults, and future investigations of identity change may consider targeting specific populations that are experiencing life transitions such as new parents, new retirees, and young people exiting secondary school and entering post-secondary education/training and the workforce.

In conclusion, we were unable to observe significant change in physical activity identity, which may have been due to a lack of change in physical activity behavior in this sample of exercise initiates. The lack of change in identity also limited our ability to examine predictors of change in physical activity identity. The findings of this investigation reinforce the strong relationship between behavioral regulation and physical activity identity, and we showed that this relationship is consistent across time. Behavioral regulation represents a potential target for future intervention research aiming to promote physical activity identity and physical activity behavior change.

## Figures and Tables

**Table 1 behavsci-14-00886-t001:** Sample demographics.

Variable	*N* = 66
Gender n (%)	
*Man*	19 (29%)
*Woman*	46 (70%)
*Missing*	1 (1%)
Minority	
*Yes*	8 (12%)
*No*	57 (86%)
*Missing*	1 (1%)
Education n (%)	
*High school diploma*	3 (5%)
*Vocational school or some college*	6 (9%)
*University/college undergraduate degree*	10 (15%)
*Professional or graduate degree*	46 (70%)
*Missing*	1 (1%)
Employment n (%)	
*Paid full-time*	24 (36%)
*Paid part-time*	19 (29%)
*Homemaker*	2 (3%)
*On leave*	1 (1%)
*Unemployed*	8 (12%)
*Retired*	2 (3%)
*Other*	7 (11%)
*Missing*	3 (5%)
Income n (%)	
*<$35,000*	16 (24%)
*$35,001–$50,000*	12 (18%)
*$50,001–$75,000*	6 (9%)
*$75,001–$100,000*	9 (14%)
*$100,001–$150,000*	8 (12%)
*>$150,000*	4 (6%)
*Chose not to respond*	8 (12%)
*Missing*	3 (5%)

Note: Income is in CAD.

**Table 2 behavsci-14-00886-t002:** Final model.

Variable	Beta Coefficient	Standard Error	*p*-Value
Intercept	2.60	0.14	<0.0001
Time	0.07	0.05	0.12
Intention	0.002	0.05	0.96
Behavioral regulation	0.26	0.05	<0.0001
MVPA	−0.0002	0.0006	0.79
ICC			
0.64			

Note: model included a random intercept. ICC = intraclass correlation coefficient.

## Data Availability

The data and R code for the main analysis can be found at OSF (https://osf.io/qu6gf/ (accessed on 10 June 2024)).

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
