# Peer review of "The Role of Intention, Behavioral Regulation, and Physical Activity Behavior in the Prediction of Physical Activity Identity across Time"

_behavsci, 2024, doi:10.3390/bs14100886_

Round 1
Reviewer 1 Report
Comments and Suggestions for Authors
I congratulate authors for this valuable research. My humble considerations are:
In title: So as to make it a bit shorter, I suggest that “Examining the” could be deleted.
In abstract: Statistics value could be deleted.
In keywords: Instead of keywords related with the design of the study, I recommend the use of others which are connected to the topic.
In introduction: Review the format in line 81 (size of the text). Which are the basis of the hypotheses? Are they based on any previous literature or based on authors’ opinion?
In methods: In 2.1. subsection, the sample is made up by 75 participants but in abstract it is said 66.
In methods: In 2.2. subsection, provide one item example of each instrument.
In discussion: I suggest to analysis in depth the practical implications of this research; its contribution to the field of knowledge it is part of.
In references: Review them; for instance, references 42 and 43 are the same.
Author Response
I congratulate authors for this valuable research. My humble considerations are:
RESPONSE: Thank you for the compliments of our research.
- In title: So as to make it a bit shorter, I suggest that “Examining the” could be deleted.
RESPONSE: Thank you for the suggestion. To make it shorter, we have now changed the title to: The role of intention, behavioral regulation, and physical activity behavior in the prediction of physical activity identity across time.
- In abstract: Statistics value could be deleted.
RESPONSE: Thank you for the suggestion, however we are going to keep the statistical information in the abstract as is the most important information to provide given these are the estimates that all of our interpretations are based on.
- In keywords: Instead of keywords related with the design of the study, I recommend the use of others which are connected to the topic.
RESPONSE: Thank you for the suggestion. We have now replaced the keyword ‘multilevel modelling’ with the keyword ‘integrated regulation.’ We are going to keep the keyword ‘longitudinal design’ as it may be relevant for searches looking for physical activity identity and longitudinal designs.
- In introduction: Review the format in line 81 (size of the text). Which are the basis of the hypotheses? Are they based on any previous literature or based on authors’ opinion?
RESPONSE: Thank you for catching the formatting issue. We will address this in the proof stage. With respect to the hypotheses, they were primarily based upon the multi-process action control (M-PAC) framework [1] and identity theories [2–5]. We now include references for the hypothesis statements on page 9 (lines 2-3 and line 5). It should be noted that we provide an overview of the M-PAC framework in the introduction, detailing why intention, behavioral regulation, and physical activity behavior should be related to physical activity identity. In the introduction, we also provided reference to identity theories specifying the importance of behavior in promoting identity development
- In methods: In 2.1. subsection, the sample is made up by 75 participants but in abstract it is said 66.
RESPONSE: The actual sample for our analysis included 66 individuals, which is why we include that number in the abstract. However, 75 individuals were originally recruited, but 66 of these individuals completed at least one measure of physical activity identity, and therefore, 66 was our effective sample size. To clarify, in the results section (page 12, line 15), we now say:
Among the 75 individuals recruited for the study, 66 participants (Mage = 34.47 years, SDage = 9.50 years) completed at least one assessment of physical activity identity and were included in the sample for this investigation.
RESPONSE:
- In methods: In 2.2. subsection, provide one item example of each instrument.
RESPONSE: Thank you for the suggestion. We now include an example item for exercise identity and behavioral regulation.
- In discussion: I suggest to analysis in depth the practical implications of this research; its contribution to the field of knowledge it is part of.
RESPONSE: Thank you for the suggestion. We now include a paragraph addressing the practical implications of the research on page 19 (lines 3-15).
- In references: Review them; for instance, references 42 and 43 are the same.
RESPONSE: Thank you for catching this, we have now merged references 42 and 43.

Reviewer 2 Report
Comments and Suggestions for Authors
The article approaches the mechanisms of physical activism which are very important for the specialists in the field. The purpose of this investigation is obvious and it was to examine whether physical activity identity develops across time among individuals new or returning to physical activity participation and whether M-PAC variables (intention, behavioral regulation, MVPA) explain significant variation in both physical activity identity across time and change in physical activity identity. Please revise the first paragraph of the Discussions subchapter, which is (almost) identical to the above-mentioned. (the last one from the page 3)
Even though the research is based on a consistent methodology, we suggest developing the analysis for the relationship between investigated variables and some demographic variables: age, gender, education level, employment, and income. It could become important for specialists in certain clubs, schools, or other facilities. Even though this idea is mentioned on page 10 (as a future direction to be investigated), why wasn’t it applied in this study? The authors seem to have all the necessary information.
The authors mentioned that It should be noted that we focused on the behavioral components of self-regulation (planning, scheduling, and self-monitoring) in this investigation. The question is: Did they notice some differences between new and more experienced participants? Maybe the former active people have learned from their teachers some rules in the area of self-regulation. Please comment on this issue and on the role of a trainer in building the process of the physical activity identity.
How did the authors motivate people to stay active? It is well known that the external assessment (external motivation), is not enough to keep the people active. Even though the authors did not intend to analyze this aspect, it could be useful for future research taking into consideration this variable.
Please explain the suggestion you mentioned that, in the future, the research should address participants who are exercise initiates with high intentions. Also, please provide recommendations for the practitioners on how to use the results in their activity.
Congratulations on your work and good luck with your future research!
Author Response
- The article approaches the mechanisms of physical activism which are very important for the specialists in the field. The purpose of this investigation is obvious and it was to examine whether physical activity identity develops across time among individuals new or returning to physical activity participation and whether M-PAC variables (intention, behavioral regulation, MVPA) explain significant variation in both physical activity identity across time and change in physical activity identity.Please revise the first paragraph of the Discussions subchapter, which is (almost) identical to the above-mentioned. (the last one from the page 3)
RESPONSE: Thank you for the suggestion. We have now reworded the first paragraph of the discussion summarizing the purpose of our study on page 15 (lines 16-17).
- Even though the research is based on a consistent methodology, we suggest developing the analysis for the relationship between investigated variables and some demographic variables: age, gender, education level, employment, and income. It could become important for specialists in certain clubs, schools, or other facilities. Even though this idea is mentioned on page 10 (as a future direction to be investigated), why wasn’t it applied in this study? The authors seem to have all the necessary information.
RESPONSE: Based on the suggestion, we examined whether age, gender, and minority status were significantly related to identity across time and change in identity (detailed on page 11, lines 17-20). Due to there being no significant relationships in any of those models, we did not include the information in the supplementary files. We also constructed an exploratory model to control for demographics, of which included a linear time slope, each M-PAC predictor (intention, behavioral regulation, and MVPA), demographic variables (age, gender, and minority status), and a random intercept (detailed on page 12, lines 1-4). We now report these results on page 15 (lines 10-14) and in Supplementary Table S7.
However, we did not conduct supplementary analyses for education level, employment, or income based on statistical/practical reasons, which include:
For education, 70% of responses are in one category, and 2 of the 4 categories have fewer than 10 responses. Therefore, there is essentially no power to detect differences between categories.
For employment, 5 of 7 categories have fewer than 10 responses, and we would only be able to compare 2 of the categories.
For income, a proper analysis would really require either (1) collecting these data on the natural continuous scale, or (2) fitting some kind of ordinal model to account for the forced discretization. (1) would not work because that's not how the data were collected (and was not a research question), and (2) does not work because such models have higher than typical sample size requirements for reasonable power; we just don't have enough data to be able to run a believable analysis for that.
Finally, from a theoretical and applied perspective, education, employment, and income have not been identified as relevant co-variates of physical activity identity in the literature.
- The authors mentioned that It should be noted that we focused on the behavioral components of self-regulation (planning, scheduling, and self-monitoring) in this investigation. The question is: Did they notice some differences between new and more experienced participants? Maybe the former active people have learned from their teachers some rules in the area of self-regulation. Please comment on this issue and on the role of a trainer in building the process of the physical activity identity.
RESPONSE: Behavioral regulation and physical activity behavior were correlated, as seen in the correlation table (Supplementary Table S2); suggesting yes, people who perform more physical activity practice more behavioral regulation. We do not discuss the correlation between behavioral-regulation and physical activity in the main text as it is beyond the scope of the paper. With respect to the role of health care professionals supporting physical activity identity, we now include a paragraph addressing the practical implications of the research on page 19 (lines 3-15), where we detail the role of health professionals supporting identity. This was also in response to reviewer 1, comment #7.
- How did the authors motivate people to stay active? It is well known that the external assessment (external motivation), is not enough to keep the people active. Even though the authors did not intend to analyze this aspect, it could be useful for future research taking into consideration this variable.
RESPSONSE: We employed an observational design, and therefore no form of intervention was used to support participants’ physical activity. Although external motivation may play a role in identity development, it is beyond the scope of our investigation and therefore we cannot speak to it in the discussion.
- Please explain the suggestion you mentioned that, in the future, the research should address participants who are exercise initiates with high intentions. Also, please provide recommendations for the practitioners on how to use the results in their activity.
RESPONSE: We believe the reviewer may be mistaken. We did not suggest future research use these samples, but rather future research is likely to include samples with new exercise initiates with high intentions. Also in relation to comment #3, we now include a paragraph addressing the practical implications of the research on page 19 (lines 3-15), where we detail the role of health professionals supporting identity
Congratulations on your work and good luck with your future research!
RESPONSE: Thank you for the kind words.
